# Constructive quantum interference in a bis-copper six-porphyrin nanoring

Sabine Richert[1,*], Jonathan Cremers[2,*], Ilya Kuprov[3], Martin D. Peeks[2], Harry L. Anderson[2] & Christiane R. Timmel[1]

The exchange interaction, $J$, between two spin centres is a convenient measure of through bond electronic communication. Here, we investigate quantum interference phenomena in a bis-copper six-porphyrin nanoring by electron paramagnetic resonance spectroscopy via measurement of the exchange coupling between the copper centres. Using an analytical expression accounting for both dipolar and exchange coupling to simulate the time traces obtained in a double electron electron resonance experiment, we demonstrate that $J$ can be quantified to high precision even in the presence of significant through-space coupling. We show that the exchange coupling between two spin centres is increased by a factor of 4.5 in the ring structure with two parallel coupling paths as compared to an otherwise identical system with just one coupling path, which is a clear signature of constructive quantum interference.

[1] Centre for Advanced Electron Spin Resonance (CAESR), Department of Chemistry, University of Oxford, South Parks Road, Oxford OX1 3QR, UK. [2] Chemistry Research Laboratory, Department of Chemistry, University of Oxford, 12 Mansfield Road, Oxford OX1 3TA, UK. [3] Department of Chemistry, University of Southampton, Highfield, Southampton SO17 1BJ, UK. *These authors contributed equally to this work. Correspondence and requests for materials should be addressed to H.L.A. (email: harry.anderson@chem.ox.ac.uk) or to C.R.T. (email: christiane.timmel@chem.ox.ac.uk).

When electrons tunnel through a molecule via more than one pathway, the wavefunctions corresponding to different routes may be in-phase or out-of-phase with each other, leading to constructive or destructive interference. Quantum interference effects of this type are thought to control electron transfer through proteins[1–3]. Quantum interference also provides many opportunities for enhancing performance in single-molecule electronic devices, by creating sharp transport resonances[4,5]. When the distance of the transmission exceeds a certain threshold, interactions with vibrational degrees of freedom result in a change of mechanism from phase-coherent single-step tunnelling to incoherent multistep hopping, erasing interference effects[6,7]. It is difficult to measure coherence lengths in molecular structures and the size limit for quantum mechanical behaviour is not well established. The simplest test for quantum coherence is to compare transmission between two points, A and B, through two identical channels in parallel, each of conductance $G_1$, with that through just one isolated channel as shown in Fig. 1.

A classical description predicts that the total conductance for the two channels will be given by Kirchhoff's circuit law as $G_{AB} = 2 G_1$; this result is expected when the distance $d_{AB}$ is greater than the coherence length. On the other hand, if the system behaves coherently, constructive quantum interference is expected to give a total conductance of $G_{AB} = 4 G_1$ (refs 4,8).

Recently, this scenario has been tested experimentally for charge transport through single molecules. STM break-junction measurements on a thioether-linked cyclophane ($d_{AB} = 0.7$ nm) gave the result $G_{AB} \approx 2.8 G_1$, providing evidence for quantum interference[9]. Similar experiments compared the conductance of a carbobenzene macrocycle ($d_{AB} = 0.8$ nm) with a single-path reference molecule to give $G_{AB} \approx 40 G_1$, although in this case the higher conductance of the two-path system is partly a consequence of its greater conformational rigidity[10]. These results highlight the challenges involved in using single-molecule charge transport measurements to probe quantum coherence.

Despite many advances in methodology[11–13], it is difficult to measure molecular conductances accurately enough to detect the predicted fourfold increase in $G_{AB}$ bestowed by constructive interference, particularly in large molecules, which are expected to be on the threshold between coherent and incoherent transport, and which have very low conductances. Furthermore, it is difficult to synthesize pairs of molecules that have identical conformations and differ only in the number of available tunnelling channels.

Here we present solutions to both these problems: We implement the scheme shown in Fig. 1 by testing the through-bond exchange coupling, $J$, between two paramagnetic centres, which can be measured accurately by electron paramagnetic resonance (EPR), and we lock the molecular wire into a well-defined conformation by using supramolecular assembly on a radial template. Comparison of the mean value of the exchange coupling between copper(II) centres in compounds **P2||P2**, with two paths, and **P2||X**, with one path gives $J_{AB} \approx 4.5 J_1$ for a through-bond tunnelling distance of $d_{AB} = 3.9$ nm, demonstrating constructive quantum interference over a remarkably long distance.

## Results

**Chemical systems.** Figure 2 shows the chemical structures of the three systems investigated in this work. The cyclic porphyrin hexamer complex on the left exhibits $D_{2h}$ symmetry with the two copper porphyrin units arranged in opposing positions, connected to each other on either side via identical zinc porphyrin dimer (**P2**) bridges. The highly conjugated structure is locked into a well-defined conformation by a hexapyridyl template which coordinates to the central metals of the porphyrin units. The ring represents a molecular analogue of the two-path model in Fig. 1 in which transmission between the two copper spin centres is possible through two identical, parallel pathways. We refer to this structure as **P2||P2**.

Interruption of the conjugation in one path of an otherwise unaltered ring assembly leads to formation of a molecular structure resembling the 'one-path' model and referred to as **P2||X**, see Fig. 2 (centre). This complex consists of a bis-copper linear porphyrin hexamer coordinated to the radial hexapyridyl template; it is extremely stable with a formation constant of more than $10^8$ M$^{-1}$ in toluene at 298 K. In the third structure, the bonds between both sets of neighbouring Zn porphyrin units have been broken and we consequently refer to this structure as **X||X**. This is the 2:1 complex formed from two equivalents of a linear porphyrin trimer and the hexapyridyl template. UV–visible–near-infrared (UV–vis–NIR) titrations show that each porphyrin trimer coordinates the template with a binding constant of $4.7 \pm 0.4 \times 10^5$ M$^{-1}$, which implies that the mole fraction of the 2:1 complex is about 0.8 in toluene at 298 K (0.2 mM). All relevant details on the synthesis, binding studies and sample preparation are given in Supplementary Figs 1–8 and Supplementary Notes 1 and 2 for **P2||X** and **X||X**. The synthesis of **P2||P2** has been reported elsewhere[14].

**Methodology.** The main focus of this work is the quantification of the interspin exchange coupling, $J$, between the two copper centres which reports on wavefunction overlap and interference phenomena. Large exchange couplings ($\gg 1$ cm$^{-1}$) can conveniently be measured by SQUID[15,16], whereas EPR can be used to determine $J$ values in the sub-cm$^{-1}$ regime, as we shall demonstrate below. The unpaired electrons also couple through space via the dipolar interaction, $D$. If EPR experiments can be conducted in the fast tumbling regime, the effects of the anisotropic dipolar coupling vanish, allowing an isolation of the isotropic exchange coupling. Such an approach was not possible here as these large molecules tumble too slowly even at room temperature.

The dipolar interaction is defined as

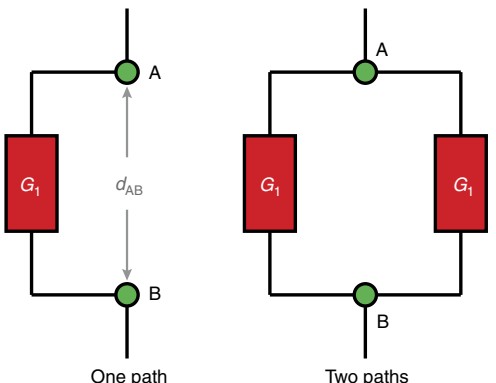

**Figure 1 | Schematic illustration of transmission through one or two paths.** Quantum coherence can be tested by comparison of the transmission properties. Assuming two identical parallel paths connecting A and B with the total conductance referred to as $G_{AB}$, coherent transmission is expected to yield $G_{AB} = 4 G_1$, whereas non-coherent transport should result in $G_{AB} = 2 G_1$. The transmission $G$ is closely related to the exchange coupling $J$.

$$D = \frac{3}{4}\frac{\mu_0}{4\pi}(g_e \beta_e)^2 \left\langle \frac{1 - 3\cos^2\theta}{r^3} \right\rangle \qquad (1)$$

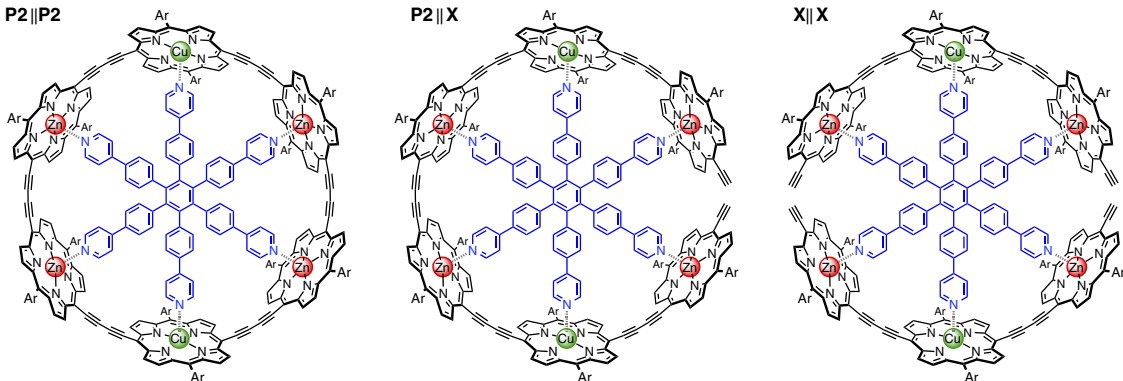

**Figure 2 | Chemical structures of the three compounds.** The structures differ in the number of pathways between the two copper centres: left—**P2∥P2**, two paths; centre—**P2∥X**, one path; right—**X∥X**, no path. The porphyrin side group 'Ar' represents an aryl substituent, 3,5-di-*tert*-butylphenyl in the case of **P2∥P2** and 3,5-bis(trihexylsilyl)phenyl for **P2∥X** and **X∥X**, which provides high solubility and prevents aggregation.

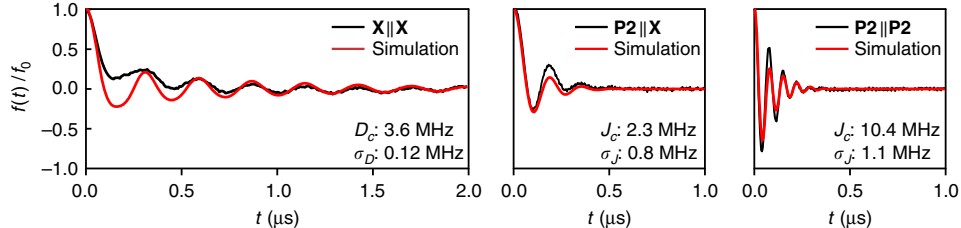

**Figure 3 | Experimental DEER data and simulations.** Black: Experimental background-corrected DEER traces of **X∥X**, **P2∥X** and **P2∥P2**. Red: Simulations of the individual experimental traces using the analytic expression given in equation (2). The simulation parameters are indicated in the graphs.

where $r$ is the interspin distance, $\theta$ is the angle between the spin-spin vector and the dipolar $Z$ axis, $g_e$ the electronic $g$-factor, $\beta_e$ the Bohr magneton and $\mu_0$ the vacuum permeability. $D$ is thus identical for all three structures whose architectures (and therefore $r$) are controlled by the same rigid template. Here we demonstrate that, with this information in hand, we can employ the technique of double electron electron resonance (DEER) to quantify and compare the exchange interaction in structures **P2∥P2** and **P2∥X**.

DEER is usually employed to elucidate molecular structures and their flexibility via the determination of interspin distances and their distributions. The latter are extracted from modulations in the DEER traces presumed to be solely due to the dipolar interaction, which, for the typically addressed interspin distances ($1.5 - 8$ nm), much exceeds the magnitude of the exchange interaction[17–19]. However, in highly $\pi$-conjugated systems, such as ours, the two couplings might be of comparable magnitude and their respective contributions to the DEER time trace need to be disentangled.

For conjugated nitroxide biradicals such a separation was successfully demonstrated for cases where both singularities of the Pake pattern could be detected after Fourier transformation of the DEER traces[20–23]. However, generally, the separation of the two contributions is difficult, especially if $D$ and $J$ are of the same order of magnitude.

To determine $J$ from the experimental DEER data, we derived the analytical expression given in equation (2), describing the dependence of the dipolar evolution (DEER) time trace on both $D$ and $J$ in the limit when the difference between the pump and probe frequencies in the absence of coupling are much larger than the pseudosecular part of the coupling Hamiltonian (weak coupling approximation).

$$f(t) = \sqrt{\tfrac{\pi}{6Dt}}\left[\cos((D+J)t)\,\mathrm{FrC}\left(\sqrt{\tfrac{6Dt}{\pi}}\right) + \sin((D+J)t)\,\mathrm{FrS}\left(\sqrt{\tfrac{6Dt}{\pi}}\right)\right] \quad (2)$$

In this equation, $f(t)$ is the reduced form factor, FrS and FrC stand for the Fresnel sine and cosine integrals, respectively, and all other parameters have their usual meanings. A full derivation of the expression is given in Supplementary Note 4. The use of this equation requires orientation selective effects on the modulation frequency to be negligible. Its applicability in the present case is justified in Supplementary Note 4.

As can be seen from the analytic result, the DEER time trace in the presence of through-bond exchange coupling has a complicated dependence on $D$ and $J$; to a first approximation, however, the experimental signal is expected to oscillate with a frequency roughly corresponding to $D + J$.

**Determination of the exchange interaction.** Equipped with an analytical expression for the DEER trace including $D$ and $J$, the first objective is the determination of the former in the absence of the latter. To this end, we quantify the dipolar coupling between the copper centres by conducting DEER on **X∥X**, the ring structure devoid of any exchange coupling paths between the copper centres.

Figure 3 (left) shows the experimental background-corrected DEER time trace recorded for **X∥X**. The frequency of the pronounced dipolar modulation corresponds to a distance of roughly 2.5 nm (at $g = 2.05$) as determined by Fourier transformation of the time trace (cf. Supplementary Fig. 10). This result is in good agreement with published X-ray crystallography and density functional theory (DFT) data for a similar template-bound six-membered porphyrin ring for which a ring diameter of 2.5 nm was determined[14,24].

A first inspection of the DEER trace for the two exchange coupled systems, **P2∥X** and **P2∥P2** in Fig. 3 (centre and right, respectively) reveals immediately that introduction of through-bond coupling paths leads to a significant increase in the modulation frequency. Fourier transformation of the traces further establishes the dominant frequency components for

**P2||X** as 5.2 MHz and for **P2||P2** as 13 MHz, providing first proof that the exchange interaction has pronounced but distinct effects in the two coupled systems. To demonstrate that the observed frequency is indeed due to intramolecular coupling rather than nuclear electron spin echo envelope modulation effects or intermolecular interactions, data for **P2||P2** were also recorded at different spectrometer frequencies and sample concentrations. The results are shown in Supplementary Fig. 9 and confirm that the high-frequency modulation indeed arises from through-bond exchange coupling since the measured modulation frequency is found to be the same in all cases.

To quantify the magnitude of $J$, the background-corrected experimental data were simulated using equation (2) assuming Gaussian distributions in the frequency domain for $D$ and $J$.

The results of the simulations are compared with the corresponding experimental data in Fig. 3. Although the relative modulation amplitudes cannot be reproduced exactly, partially due to uncertainties in the DEER background correction, the simulations can be considered satisfactory since the modulation frequencies and the dampening of the oscillations, which contain all the relevant information for our purposes, can be well reproduced and determined with high precision. The background signal in the DEER traces can be attributed to incomplete complex formation, as discussed in Supplementary Note 5 and Supplementary Figs 13–15. The conclusions of the discussion of the background signal are also supported by electron-nuclear double resonance measurements shown in Supplementary Fig. 11 and described in Supplementary Note 3.

The centres and widths of the frequency distributions in $J$ and $D$, obtained from the simulations, are indicated in the corresponding graphs in Fig. 3. First, the distribution in $D$ was determined using the **X||X** sample with two breaks in the conjugation. The resulting distribution then served as an input into the simulations of the **P2||X** and **X||X** DEER traces in which, consequently, only the centre and width of the distribution in $J$ were adapted. The distribution in $D$ was found to be relatively narrow corresponding to a width of $\sigma = 0.12$ MHz, whereas the distributions in $J$ are generally found to be much wider. The centre frequency of 3.6 MHz determined for $D$ corresponds to a Cu⋯Cu distance of 2.47 nm at $g = 2.05$, in excellent agreement with expectations from available experimental data and DFT calculations[14,24].

From equation (2) it appears that only the relative sign of $D$ and $J$ can in principle be determined by a simulation of the experimental data using this relation. However, it has been shown that the frequency distribution of combined dipole–dipole and exchange coupling does depend on the sign of the exchange coupling[17], which enables us to determine the absolute value of $J$ as also confirmed in Supplementary Fig. 12 and discussed in Supplementary Note 5. The sign of $J$ could also be confirmed by DFT calculations as described in Supplementary Note 6.

For **P2||P2**, an exchange coupling of 10.4 MHz ($\sigma_J = 1.1$ MHz) could be determined from the simulation. Since equation (2) was derived for an exchange coupling Hamiltonian of $\hat{H}_J = 2\pi J\, S_A S_B$, a positive $J$ value indicates antiferromagnetic coupling ($E_S < E_T$). The magnitude of the exchange interaction between the two copper centres is remarkable given the large through-bond interspin distance of 3.9 nm, certainly a consequence of the high degree of conjugation between the paramagnetic centres[25].

The most crucial finding, however, is that interruption of one coupling path in **P2||X** leads to attenuation of $J$ to just 2.3 MHz.

## Discussion

We observe an approximately fourfold increase in the size of the exchange coupling in the two-path structure as compared to the one-path case. This result is in excellent agreement with the prediction of a fourfold increase in the transmission of a closed system with two parallel branches as compared to a single linear chain under conditions of constructive quantum interference[4]. Although quantum interference has been observed in two-path systems by conductance measurements[9,10], this is the first time that it has been detected via exchange coupling.

The major advantage of employing DEER is that the data allow the direct quantification of the exchange coupling between the spins alleviating the need for any connecting electrodes. Our study does not report on conductance phenomena directly but rather addresses quantum interference via the exchange interaction between the two paramagnetic centres. However, it has been shown previously that conductance and antiferromagnetic coupling exhibit strongly correlated trends[26,27] so that the fourfold increase in the exchange coupling by introduction of a second, parallel path is consistent with the prediction based on transmission considerations.

These results provide an unambiguous demonstration of constructive quantum interference in a closed model system with two identical parallel paths. They illustrate the applicability of DEER for measuring long-range exchange couplings in molecular wires and indicate that efficient long-range electronic communication is favoured by a highly rigid molecular framework.

The nanorings studied here provide a rich playground for further work. The effects of inequivalent pathways (arranging the copper centres in *ortho* or *meta* positions), ring size, connecting geometry and linkers are presently being tested in our laboratory.

## Methods

**Sample preparation.** The synthesis of the compounds used in this study is presented in Supplementary Note 1. UV–vis–NIR titrations shown in Supplementary Figs 5–8 and discussed in Supplementary Note 2 demonstrate formation of the complexes shown in Fig. 2. For the preparation of the EPR samples, the porphyrin oligomers were first dissolved in $CHCl_3$. The hexapyridyl template was added in stoichiometric amounts for complex formation (0.5 eq. for **X||X**, 1 eq. for **P2||X**) and complex formation was verified by UV–vis–NIR spectroscopy. The solvent was then removed under high vacuum conditions and the complex subsequently redissolved in deuterated toluene to yield a solution with a concentration of about 0.2 mM. After verification of complex formation in toluene, the solutions were degassed by the freeze–pump–thaw method and backfilled with argon. The deoxygenated solutions were then transferred to an EPR tube and the EPR tube closed with a subaseal. For storage in liquid nitrogen the subaseal was removed. The frozen samples were directly inserted into the EPR resonator for the measurements at 15 K.

**Details of the EPR measurements.** DEER experiments were performed at 15 K and Q-band frequencies on a Bruker ELEXSYS E580 spectrometer equipped with a Bruker EN 5107D2 resonator and a liquid helium flow cryostat using the sequence $\pi/2 - \tau_1 - \pi - \tau_1 - \tau_2 - \pi - \tau_2$-echo at the detection frequency $\nu_{det}$ while applying a single $\pi$ pulse at the pump frequency $\nu_{pump}$ during the interval $\tau_1 - \tau_2$. The pump $\pi$ pulse position was varied step-wise starting at a time $t_0 < \tau_1$ after the first detection $\pi$ pulse up to a time $t < \tau_1 + \tau_2$, shortly before the second $\pi$ pulse at $\nu_{det}$.

A pulse length of 12 ns was chosen for the pump pulse, whereas all detection pulses had a length of 16 ns. A 16-step phase cycle needed to be employed to remove unwanted echos (arbitrary waveform generator (AWG), coherent source). Deuterium nuclear modulations were averaged by increasing $\tau_1$ in eight steps of 16 ns, typically starting at $\tau_1 = 400$ ns. To record the traces shown in the text, the pump frequency was set to the maximum of the field-swept EPR spectrum ($xy$) and the detection frequency was $\nu_{det} = \nu_{pump} + 100$ MHz. The pump pulse position was varied in steps of either 4 or 8 ns and data were collected at a repetition rate of 2 μs using $\tau_2$ values between 2 and 3 μs.

**Data availability.** The data that support the findings of this study are available from the corresponding authors on reasonable request.

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

## Acknowledgements

Financial support from the EPSRC (EPL011972/1, EP/M016110/1 and EP/J015067/1) and the ERC (Grant 320969) is gratefully acknowledged. We thank the EPSRC UK National Mass Spectrometry Facility at Swansea University for mass spectra. We would like to acknowledge the use of the University of Oxford Advanced Research Computing (ARC) facility in carrying out this work (http://dx.doi.org/10.5281/zenodo.22558).

## Author contributions

J.C. synthesized (BE) the compounds and demonstrated formation of the supramolecular complexes. S.R. performed the EPR experiments and analysed the data. S.R. and I.K. performed the simulations. M.D.P. carried out DFT calculations. C.R.T. and H.L.A. coordinated the study. S.R., H.L.A. and C.R.T. wrote the manuscript. All authors discussed the results and commented on the manuscript.

## Additional information

**Competing interests:** The authors declare no competing financial interests.

**Publisher's note**: 

