## [Peer Review File · Nature Communications]

Reviewers' comments:

Reviewer #1 (Remarks to the Author):

This manuscript reports through bond long range (3.9 nm) electronic communication controlled by quantum coherence. The studies presented in the manuscript are unique, going along with a good molecular design to study the electronic effects. The paper can be accepted for publication.

There are however a few comments:

1. It would be perhaps more informative if coupling effects could be compared with different topologies of the electronic path (linear vs circular, which would be the P2//X molecule, without and with the template).
2. How is the coupling pattern in case of molecule P2//P2 without the template? Does that have significant differences with templates P2//P2?

Reviewer #2 (Remarks to the Author):

In this manuscript, the authors proposed and demonstrated a new way of measuring quantum interference in pi-conjugated structures through electron paramagnetic resonance spectroscopy. In the setup, six porphyrin rings coordinated to zinc or copper are anchored to a hexapyridyl template in a circle with two copper paramagnetic centers arranged in opposite positions. By connecting the porphyrin rings with alkyne linkers in different ways, zero to two pathways are created between the two copper centers. After that, EPR spectrum is measured. With the analytical expression of the DEER time trace as a function of the through-bond exchange coupling J and through-space dipolar coupling D , the magnitude of J is quantitatively determined by combing the experimental data with the simulation results. The authors observed that in the two pathways layout, J is about four times the value of that in single path case, which indicates the occurrence of constructive interference.

This manuscript is well written and the works are done neat and clean. Measuring the exchange coupling strength between paramagnetic centers instead of the conductance in molecular junctions indeed provides us an alternative tool in the research of quantum interference.

The only question I have is that, although the way in which the six porphyrin rings are fixed is very clever and effective, will the exchange coupling J be less if the structure becomes more flexible or the temperature becomes higher, considering the dephasing effect due to nuclear vibrations.

On the whole, I recommend the article to be published.

Reviewer #3 (Remarks to the Author):

This manuscript experimentally demonstrates for the first time constructive quantum interference of two exchange coupling pathways by a precise measurement of the exchange coupling in two rigid molecules with virtually the same geometry and one or two equivalent exchange pathways. This result is related to constructive quantum interference in electric conduction and, potentially, in optimized electron transfer in biological systems and thus of high general interest. The work has been performed very carefully and the conclusions are fully supported by the experimental evidence and data analysis. The manuscript is generally well written, concise, and clear. A few issues require minor revision. After such revision, I strongly support publication in Nature Communications.

Details:

1. Equation (2) is valid only in the weak coupling limit, where the difference between the resonance frequencies of the two spins in the absence of coupling are much larger than the pseudo-secular part of the coupling Hamiltonian. This information is missing both in the main text and in the Supporting Information. The approximation is valid in the case at hand, as the difference between observer and pump frequency ensures this condition, but it is important to alert the reader to the use of this approximation.

2. I take issue with the footnote on page 5. Both detection and pump pulses select magnetic field directions near the respective porphyrin (pseudo)planes. It follows that the magnetic field vector is always close to perpendicular to the spin-spin vector. Orientations near $\theta = 0$ are suppressed. This needs to be acknowledged and may contribute to the slight disagreement between experiment and simulations (which does not affect the conclusions).

3. I believe that the footnote on p. 6 is not required. It is known (see, for instance, Ref. 17) that the frequency distribution of combined dipole-dipole and exchange coupling does depend on the sign of the exchange coupling. As the spectra are different, the time-domain data must be different. Orientation selection effects could be that strong as to hide the difference and for this reason the test in the Supporting Information is useful. However, the reasoning should be clearer.

4. Typo, page S10, line 4 below figures and captions: 'probably' should read 'probability'.

5. Between Eq. (S2,S3) and Eq. (S5) the Hamiltonian is truncated (high-field and weak coupling approximations). This needs to be mentioned. If the Hamiltonian could not be truncated, than removal of the Zeeman Hamiltonian (which is refocused by the observer echo sequence) would not be permissible.

6. I believe that the remaining difference between experimental data and simulation in Fig. S13 may be a consequence of orientation selection.

7. It is somewhat unsatisfying that no DFT computation was performed for P2||X. Unless this is done, the DFT computation for P2||P2 does not tell us much.

We are grateful for the opportunity to reply to the reviewers' comments on the above named manuscript and are delighted that they agree that we present interesting research in a well-written paper. The few minor changes requested have now been addressed as shown below.

Responses to reviewer 1:

- 1. It would be perhaps more informative if coupling effects could be compared with different topologies of the electronic path (linear vs. circular, which would be the P2||X molecule, without and with the template).*

The comparison of the DEER data for P2||X with and without template is presented in the Supplementary Information (Supplementary Figure 15). We agree that this comparison is very informative and now added an additional paragraph in the Supplementary Information (Supplementary Note 5) discussing the effect of the molecular template, rigidifying the geometry. No effects of J-coupling could be observed in the DEER data for the 'linear' compound, most likely due to a broad distribution in intra-molecular couplings owing to the increased flexibility.

- 2. How is the coupling pattern in case of molecule P2||P2 without the template? Does that have significant differences with templates P2||P2?*

We agree that the comparison between P2||P2 with and without template might be interesting although the associated loss of the rigidity of the structure seems likely to have a detrimental effect on our ability to observe any J-coupling in DEER time traces, as explained in our answer above to point 1 and in reply to the question raised by referee 2. These measurements are hence unlikely to contribute to any discussion on quantum interference effects.

Moreover, the synthetic effort required to produce this compound in sufficient quantity and purity for EPR experiments would probably postpone publication of this article by several months: the compound is synthesised via the template-bound P2|P2 structure (using template-directed coupling). The synthesis of the latter is challenging and the compound can only be obtained in low yields (cf. Reference 14 of the main text). The binding to the template is extremely strong and we have not yet developed a practical method for removing the template to prepare the template-free P2|P2 compound (cf. Supplementary Note 2).

Responses to reviewer 2:

1. *The only question I have is that, although the way in which the six porphyrin rings are fixed is very clever and effective, will the exchange coupling J be less if the structure becomes more flexible or the temperature becomes higher, considering the dephasing effect due to nuclear vibrations?*

As already explained above, from the measurements undertaken in this study, we can conclude that the rigidity of the structure is indeed crucial for the observation of J-coupling in DEER time traces (see above). For example, in Supplementary Figure 15, we present the comparison between the DEER traces of P2|X with and without template. Without the template imposing a rigid, curved geometry, the molecule is likely to be rather 'linear' (porphyrin chains are inherently very flexible), with a considerably increased flexibility. The comparison of the DEER data reveals that no distinct J-coupling frequency can be observed under these conditions, presumably due to a very broad distribution of intra-molecular couplings. A paragraph has now been added to the Supplementary Information discussing the importance of the rigidity of the structure based on Supplementary Figure 15.

We expect that raising the temperature would have a considerable effect on the observed DEER traces since the distribution of couplings depends on kT and should become broader at higher temperatures due to the increased structural flexibility. Unfortunately we are not able to perform these measurements at higher temperatures since the feasibility of the experiment depends on the transverse relaxation time. We require the latter to be sufficiently long, which limits the accessible temperature range to about < 40 K for the studied copper-containing molecules. The molecules are frozen under these conditions and the distribution of conformations should in all cases correspond to that at the freezing point of the solvent (toluene) of about 180 K.

Responses to reviewer 3:

1. *Equation (2) is valid only in the weak coupling limit, where the difference between the resonance frequencies of the two spins in the absence of coupling are much larger than the pseudo-secular part of the coupling Hamiltonian. This information is missing both in the main text and in the Supporting Information. The approximation is valid in the case at hand, as the difference between observer and pump frequency ensures this condition, but it is important to alert the reader to the use of this approximation.*

We are grateful to the reviewer for prompting us to include this information and we have modified the sentence in the main text accordingly. It now reads "To determine J from the experimental DEER data, we derived the analytical expression given in Eq. (2), describing the dependence of the dipolar evolution (DEER) time trace on both D and J in the limit when the difference between the pump and probe frequencies in the absence of coupling are much larger

than the pseudo-secular part of the coupling Hamiltonian (weak coupling approximation)."

We also added this information to the derivation in the Supporting Information as specified below under point 5.

- 2. I take issue with the footnote on page 5. Both detection and pump pulses select magnetic field directions near the respective porphyrin (pseudo)planes. It follows that the magnetic field vector is always close to perpendicular to the spin-spin vector. Orientations near $\theta=0$ are suppressed. This needs to be acknowledged and may contribute to the slight disagreement between experiment and simulations (which does not affect the conclusions).*

We agree with the reviewer that our formulation of the footnote on page 5 was probably misleading. The footnote was now modified to include the points raised by the reviewer.

- 3. I believe that the footnote on p.6 is not required. It is known (see, for instance, Ref. 17) that the frequency distribution of combined dipole-dipole and exchange coupling does depend on the sign of the exchange coupling. As the spectra are different, the time-domain data must be different. Orientation selection effects could be that strong as to hide the difference and for this reason the test in the Supporting Information is useful. However, the reasoning should be clearer.*

We thank the reviewer for this comment. The footnote was removed and the reasoning the main text adapted accordingly.

- 4. Typo, page S10, line 4 below figures and captions: 'probably' should read 'probability'*

The typo in the Supplementary Information has been removed.

- 5. Between Eq. (S2,S3) and Eq. (S5) the Hamiltonian is truncated (high field and weak coupling approximations). This needs to be mentioned. If the Hamiltonian could not be truncated, then removal of the Zeeman Hamiltonian (which is refocused by the observer echo sequence) would not be permissible.*

A sentence has now been added in the Supporting Information between Eq. (S2,S3) and Eq. (S5) detailing the approximations made in the derivation of the analytic expression.

- 6. I believe that the remaining difference between experimental data and simulation in Fig. S13 may be a consequence of orientation selection.*

We agree that this might be the case and added a sentence in the Supplementary Information mentioning this.

- 7. It is somewhat unsatisfying that no DFT computation was performed for $P2||X$. Unless this is done, the DFT computation for $P2||P2$ does not tell us much.*

We agree that no conclusions on quantum interference can be drawn solely from the DFT calculation for $P2||P2$. Originally this calculation was only performed and included to confirm the sign of the exchange interaction found experimentally. As detailed in the literature, the calculation

of J-couplings as small as those in this work is at the limit of the capabilities of DFT. It is therefore to be expected that the magnitudes are unreliable and error-prone. We were very careful not to over-interpret the obtained values and therefore refrained from a direct comparison of experimental data and DFT. Nevertheless, to give the reader an idea of the spread of J-values that are obtained by DFT for the studied molecules, additional computational results have now been included in the Supplementary Information for a model of P2||X. Details of the computational analysis using two different methods are given in Supplementary Note 6.

REVIEWERS' COMMENTS:

Reviewer #1 (Remarks to the Author):

I am satisfied with the explanations and additions in response to my questions. I recommend strongly the acceptance.

Reviewer #2 (Remarks to the Author):

The modifications made in the supplementary information is convincing and I recommend the paper for publication.

Reviewer #3 (Remarks to the Author):

The authors have revised their manuscript to my complete satisfaction. I recommend publication in Nature Communications in the present form.

Authors' reply

We thank all reviewers again for their comments and suggestions. Since the reviewers' request no further changes, the scientific content of the manuscript was not modified since the last revision.